**Near Real Time Arctic sea ice thickness and volume from CryoSat-2**

R. L. Tilling[1], A. Ridout, A. Shepherd[2]

[1] Centre for Polar Observation and Modelling, Department of Earth Sciences, University College London, London, WC1E 6BT, UK

[2] Centre for Polar Observation and Modelling, School of Earth and Environment, University of Leeds, Leeds, LS2 9JT, UK

*Correspondence to*: R. L. Tilling (rachel.tilling.12@ucl.ac.uk)

**Abstract.** Timely observations of sea ice thickness help us to understand Arctic climate, and can support maritime activities in the Polar Regions. Although it is possible to calculate Arctic sea ice thickness using measurements acquired by CryoSat-2, the latency of the final release dataset is typically one month, due to the time required to determine precise satellite orbits. We use a new fast delivery CryoSat-2 dataset based on preliminary orbits to compute Arctic sea ice thickness in near real time (NRT), and analyse this data for one sea ice growth season from October 2014 to April 2015. We show that this NRT sea ice thickness product is of comparable accuracy to that produced using the final release CryoSat-2 data, with a mean thickness difference of 0.9 cm, demonstrating that the satellite orbit is not a critical factor in determining sea ice freeboard. In addition, the CryoSat-2 fast delivery product also provides measurements of Arctic sea ice thickness within three days of acquisition by the satellite, and a measurement is delivered, on average, within 14, 7 and 6 km of each location in the Arctic every 2, 14 and 28 days respectively. The CryoSat-2 NRT sea ice thickness dataset provides an additional constraint for seasonal predictions of Arctic climate change, and will allow industries such as tourism and transport to navigate the polar oceans with safety and care.

## 1      Introduction

Arctic sea ice is a key component of the global climate system, and changes in its thickness and volume impact on regional heat (Sedlar et al., 2011) and freshwater (Aagaard and Carmack, 1989) budgets, and on subsequent patterns of atmospheric (Singarayer et al., 2006, Schweiger et al., 2008, Francis and Vavrus, 2012) and oceanic (Vellinga and Wood, 2002) circulation across the Arctic and at lower latitudes. The availability of Arctic-wide sea ice thickness data, especially in near real time (NRT), will enable evaluation and improved skill in the prediction of sea ice thickness distributions by climate models (Day et al., 2014) which, in turn, will benefit models of the global climate. In addition, there is increasing interest in the behaviour of Arctic sea ice among operational services, with a growing need for accurate and timely information of sea ice thickness. For example, shipping through the Arctic Ocean via the Northern Sea Route (NSR) could save about 40% of the sailing distance from Asia (Yokohama) to Europe (Rotterdam) compared to the traditional route via the Suez Canal (Liu and Kronbak, 2010), which would quicken the regional export of natural resources, and delivery of cargo to the communities along the Siberian coast (Meier et al., 2014). Ease of passage is also a concern for those looking to ship along the Northwest Passage and future trans-Arctic shipping routes along the Russian coast, and when considering the potential for tourism in regions such as Canadian Arctic waters (Stewart et al., 2007). The oil and gas sector require hemispheric studies of sea ice concentration, extent, motion and thickness (Galley et al., 2013) to estimate productions costs and to assess the feasibility and safety of replacing ice-based construction with lower cost conventional construction equipment (Harsem et al., 2011). Without these studies, companies cannot be sure that their infrastructure is suitably robust for the Arctic environment, such as when the Shell oil rig Kulluk ran aground in January 2013. As a consequence many large oil companies are reducing their plans for Arctic exploration and drilling activities due to the high costs and risks and the possibility of safer investment in other regions. This will impact on northern areas and communities through local businesses who report losses in hotel revenues, restaurant businesses, and the local marine support (Meier et al., 2014). Up-to-date measurements of sea ice thickness are crucial when considering building costs for exploration platforms and icebreaker ships, transit speeds, and navigation difficulties and risks. Here we present a method for obtaining NRT sea ice thickness measurements across the northern hemisphere using fast delivery CryoSat-2 data.

A range of Arctic sea ice thickness measurements are currently available, with varying spatial and temporal coverage. The Beaufort Gyre Exploration Project (BGEP), based at the Woods Hole Oceanographic Institution in collaboration with researchers from Fisheries and Oceans Canada at the Institute of Ocean Sciences, have provided year-round sea ice draft data from upward

looking sonar buoys since 2003, from three buoys in the Beaufort Sea. On a larger scale, NASA's Operation IceBridge utilises a suite of research aircraft each spring (March and April) to produce tracks of sea ice thickness estimates (Kurtz et al., 2013) concentrated around northern Greenland, the ocean region north of the Canadian Archipelago, and the Beaufort Sea. Currently the final and 'quick look' IceBridge data are available for spring 2009-2012 and spring 2013-2015, respectively. The quick look product is experimental and is designed only to be applicable for time-sensitive projects such as sea ice forecasting. On a larger spatial scale, there are currently three publically available datasets that provide sea ice thickness estimates across the whole Arctic Ocean. These datasets are produced by NASA (Kurtz et al., 2014), Germany's Alfred Wegener Institute (AWI) (Ricker et al., 2014), and the UK's Centre for Polar Observation and Modelling (CPOM) (Tilling et al., 2015) using final release data from the European Space Agency's (ESA) CryoSat-2 satellite (Wingham et al., 2006), which was launched in 2010. NASA provide monthly-averaged thickness data for March 2014 and March 2015 within a fixed central Arctic region that covers an area of ~$7.2\times10^6$ km$^2$. The region was first defined for use with the NASA ICESat satellite (Kwok et al., 2009), and will hereafter be referred to as the ICESat domain. The NASA product is currently quick-look and experimental. AWI provide monthly averaged thickness data starting from January 2011 with a current lag of about 6 months, and these data again cover a central area of the Arctic Ocean. CPOM distribute sea ice thickness estimates for spring (March/April average) and autumn (October/November average) beginning in autumn 2010, also with a lag of about 6 months, depending on the availability of sea ice concentration data (Cavalieri et al., 1996, updated yearly). The CPOM estimates cover the entire northern hemisphere, defined as latitudes above and including 40° N.

## 2    Data and Methods

We use fast delivery radar altimeter measurements from the ESA CryoSat-2 satellite synthetic aperture radar (SAR) and SAR interferometric (SARIn) altimeter (Wingham et al., 2006) to produce NRT estimates of Northern Hemisphere (latitudes above 40° N) sea ice thickness and volume. The data are Level 1b, and consist of an echo for each point along the ground track of the satellite. For Arctic sea ice processing we assume that the ice surface is relatively flat and that slope variations are minimal. Under these circumstances, echoes are received primarily from the nadir point beneath the satellite ground track. We crop the SARIn mode waveforms to include only the central 128 range bins to allow for identical processing of SAR and SARIn mode data as both now have 128 bins in their waveform data. Prior to the release of Level 1b data, ESA perform some on-ground processing of the raw satellite data. Before March 26th 2015, ESA applied a processing chain known as 'Baseline-B' to the raw fast delivery data, and an updated processor, 'Baseline-C', has been applied since.

In the fast delivery data the wet tropospheric, dry tropospheric and inverse barometer corrections are missing in 94% of cases for Baseline-B data, but in less than 1% of cases for Baseline-C data. In these instances, all three of the corrections are missing. The fast delivery CryoSat-2 data are available from ESA on average 36 hours after acquisition by the satellite, although we run our sea ice processor with a latency of three days to ensure sufficient data are available. The main difference between the fast delivery and final release CryoSat-2 data is the orbits applied. For both datasets, an accurate determination of the satellite orbit is required to determine surface elevations above a reference ellipsoid. For the final release data product, ESA perform a ground-based Precise Orbit Determination (POD), which requires modelling of the forces acting on the satellite as well as a dense set of measurements regarding its position and velocity (Wingham et al., 2006). The primary means of making these measurements is with the on-board Doppler Orbit and Radio positioning Integration by Satellite (DORIS) receiver, which makes measurements of the relative velocity of the satellite to an extensive network of ground beacons. The messages uplinked from the beacons include time signals that allow the DORIS receiver time to be accurately determined. The DORIS receiver also includes software for the real-time, on-board computation of the orbit, known as the DORIS Navigator orbit. The DORIS

Navigator orbit is estimated to be accurate to 30 cm in the radial direction, and is included in the fast delivery CryoSat-2 data to provide good quality orbit estimates before the POD can be produced. However, the fast delivery data are more susceptible to orbit dropout, meaning that certain orbits, for which the orientation of the satellite could not be sufficiently determined, are not included in the dataset. There is also a difference in the timeframe of on-ground processing of the raw fast delivery and final release data by ESA. Before February 22nd 2015, ESA applied the Baseline-B processing chain to the raw final release data, and an updated processor, Baseline-C, has been applied since April 1st 2015. Between these dates, a hybrid processor known as 'Baseline-BC' was applied. On average, it takes us six hours to process one day of data.

The processing steps for fast delivery CryoSat-2 data are identical to those used for the final delivery data, and are described in Tilling *et al.* (2015).  The first step is the computation of sea ice freeboard, which is the difference in elevation between the snow-ice interface and that of the surrounding ocean. We do this by using the return echo shape to discriminate between measurements of the ocean surface and the ice surface (Peacock and Laxon, 2004). We define sea ice regions as those with a NRT sea ice concentration (Maslanik and Stroeve, 1999, updated daily) greater than 75%. NRT ice concentration data are taken from the National Snow and Ice Data Center (NSIDC) and are available to us by 01:00 UTC two days after measurement. A correction is applied to each freeboard measurement to account for the reduced speed of the radar pulse as it passes through any snow cover on sea ice. The next step is to convert sea ice freeboard to sea ice thickness. We assume that the ice floes are in hydrostatic equilibrium, under which circumstances sea ice thickness can be calculated using:

$$T_i = \frac{f_c \rho_w + h_s \rho_s}{\rho_w - \rho_i} \tag{1}$$

where $T_i$ is the sea ice thickness, $f_c$ is the corrected sea ice freeboard, $h_s$ is snow depth, $\rho_w$ is seawater density, $\rho_s$ is snow density, and $\rho_i$ is sea ice density. We use a fixed estimate of first-year ice (FYI) density of 916.7 kg m$^{-3}$ (Alexandrov et al., 2010), multi-year ice (MYI) density of 882 kg m$^{-3}$ (Alexandrov et al., 2010), and a fixed seawater density of 1,023.9 kg m$^{-3}$ (Wadhams et al., 1992). To obtain snow depth and density we average the values from a climatology (Warren et al., 1999) that fall within the ICESat domain, where the climatology is constrained by *in situ* measurements. Snow depth is halved over FYI to account for reduced snow accumulation (Kurtz and Farrell, 2011, Webster et al., 2014). NRT ice type data from the Norwegian Meteorological Service Ocean and Sea Ice Satellite Application Facility (http://osisaf.met.no/p/ice/#type) are used to classify FYI and MYI for each individual freeboard measurement, and this dataset is available to us by 01:00 UTC the day after measurement. During the sea ice melt season it becomes difficult to discriminate between measurements of the ocean and the ice due to melt ponds that form on the sea ice surface, and because of this we do not currently produce measurements of sea ice thickness between May and September. NRT sea ice thickness data are output Arctic-wide on a 5 km square grid (Fig. 1), or for user-configurable regions of interest (ROI) on a 1 km square grid. To obtain Arctic-wide and ROI grid values, we average all thickness measurements within a 25 and 5 km radius of the centre of the grid, respectively, with all points receiving equal weighting. We then compute sea ice volume Arctic-wide and within fixed oceanographic basins (Nurser and Bacon, 2014, Tilling et al., 2015) by averaging individual thickness and concentration values during each calendar month on a 0.1 by 0.5 degree grid, and defining the sea ice margin by applying a 15% sea ice concentration mask using data from the 15th day of each month. Empty thickness grid cells within the sea ice extent mask, including those north of 88°N, are filled by nearest neighbour interpolation with a maximum search radius of 300 km. Monthly estimates of sea ice volume are then calculated by summing the product of the ice thickness, the ice concentration, and the ice

area, within the sea ice extent mask.

We estimate monthly errors in sea ice volume (Tilling et al., 2015) by considering the contributions due to uncertainties in sea ice freeboard (~9 cm), snow depth (4.0 to 6.2 cm in Warren et al., 1999), snow density (60.0 to 81.6 kg m$^{-3}$ in (Warren et al., 1999), sea ice density

(7.6 kg m$^{-3}$ in(Romanov, 2004, Tilling et al., 2015),  sea ice concentration (5% according to the NSIDC at http://nsidc.org/data/docs/daac/nsidc0051_gsfc_seaice.gd.html), sea ice extent (20,000 to 30,000 km$^2$ according to the NSIDC at http://nsidc.org/arcticseaicenews/faq/#error_bars), and sea ice freeboard. Uncertainties in seawater density are neglected because they have a negligible impact (Kurtz et al., 2013, Ricker

et al., 2014).

Errors in our freeboard estimates arise through speckle in the radar echoes, which averages 8 cm across the Arctic but de-correlates from one measurement to the next, and from uncertainties in sea surface height, which may be correlated in space due to our interpolation scheme based on a linear regression of measurements along 200 km sections of the ground

track. We examined the variability of sea surface heights over this scale, and their standard deviation at orbit crossing points is 4 cm. As a conservative estimate, we assume that this variability remains correlated within the 200 km window of our freeboard calculation, and include it as an additional source of uncertainty in our gridded product. The freeboard error is then a combination of that due to spatially uncorrelated speckle on floe heights and that due to

spatially correlated errors in the interpolation of sea surface heights. This results in a 2 cm freeboard uncertainty, which scales to ~20 cm thickness, or 11% of a typical growth season thickness of 1.8 m (Tilling et al., 2015) for our gridded, 28-day product.

To calculate uncertainties in sea ice volume, we compute the monthly rate of change of volume with respect to each parameter that has an associated error. We do this by individually

adjusting the value for each parameter six times, at even increments, and re-computing the volume each time. The computed rates of change are then multiplied by the error in each parameter in question to estimate their partial contributions to the total volume error. Finally, we combine the monthly contribution to the volume error for all significant error sources in a root-sum-square manner to arrive at an estimate of the total monthly sea ice volume error,

using:

$$\sigma_V = \sqrt{\left(\frac{\partial V}{\partial h_s} \cdot \sigma_{h_s}\right)^2 + \left(\frac{\partial V}{\partial \rho_s} \cdot \sigma_{\rho_s}\right)^2 + \left(\frac{\partial V}{\partial \rho_i} \cdot \sigma_{\rho_i}\right)^2 + \left(\frac{\partial V}{\partial e_i} \cdot \sigma_{e_i}\right)^2 + {\sigma_{V_c}}^2} \qquad (2)$$

where $\sigma_V$ is the uncertainty in sea ice volume in a given month, $V$ is sea ice volume, $h_s$ is Arctic-wide snow depth, $\sigma_{h_s}$ is the uncertainty in snow depth, $\rho_s$ is Arctic-wide snow density, $\sigma_{\rho_s}$ is the uncertainty in snow density, $\rho_i$ is Arctic-wide ice density, $\sigma_{\rho_i}$ is the uncertainty in sea ice

density, $e_i$ is sea ice extent, $\sigma_{e_i}$ is the uncertainty in sea ice extent, and $\sigma_{V_c}$ is the uncertainty in sea ice volume due to uncertainty in sea ice concentration. We estimate that year-to-year uncertainties in Arctic-wide sea ice volume are typically about 13.5%, with small variations from month to month (Tilling et al., 2015).

Estimating local errors in sea ice thickness is complicated due to a lack of knowledge of the

distances over which the contributing factors de-correlate. The main factors for which this information is important and lacking are snow depth, snow density, and sea ice density. In our sea ice volume error budget, we estimate their uncertainty over large scales as the standard deviation of monthly-averaged sparse field observations collected across the 9 million km$^2$ central Arctic region. However, these factors, and their variability, are influenced by synoptic-

scale meteorology, and we suppose that the length scale over which they are correlated is

comparable to that of a typical polar vortex – around 2000 km in diameter (http://www.cpc.ncep.noaa.gov/products/stratosphere/polar/ polar.shtml). Taking snow depth as an example, over areas that are large in comparison to this correlation scale, the variability of spatially averaged snowfall fluctuations will diminish in the ratio $1/\sqrt{n}$, where $n$ is the effective number of independent values of accumulation sampled. We take $n \sim A/(\pi 2000^2)$, where $A$ is the area in square kilometres. If $n < 1$, we set it equal to 1. For the 9 million km² central Arctic region, over which the large scale sea ice volume and thickness uncertainty is estimated to be 13.5%, $n \sim 3$, leading to an uncertainty of 23%. Using this approach, and accounting additionally for short-scale correlated errors in freeboard associated with interpolating sea surface heights, we estimate the uncertainty in sea ice thickness increases to 25% at the 5 km scale of our 28-day NRT grid.

We acknowledge that this is only a first attempt to characterise local uncertainty in sea ice thickness, and that more detailed observations of snow depth, snow density, and sea ice density are required to establish the extent to which their variability impacts on the retrieval accuracy. However, a 23% local error in our gridded, 28-day estimates of Arctic sea ice thickness derived from CryoSat-2 observations corresponds to an uncertainty of 41 cm for a typical thickness of 1.8 m. This uncertainty is consistent with the spread of differences relative to independent estimates acquired from airborne and ocean-based platforms (34 to 66 cm in Tilling et al., 2015). However, grid cell thickness uncertainty will increase with fewer days of data coverage. For example, for 2 days of data the averaged freeboard measurements often come from just one satellite pass. Therefore the full 4 cm uncertainty in sea surface height contributes to the freeboard error, which scales to ~40 cm for thickness, or 22% of a typical thickness of 1.8 m. Combined with the error of 23% from other sources this brings the total error on the 2 day 5 km grid sea ice thickness data to 32%.

To assess the reliability of our NRT sea ice data set we compared it to values derived from the final CryoSat-2 data release (the archive product), which have shown excellent agreement with an extensive set of independent observations (Tilling et al., 2015). It is currently not possible to evaluate our NRT sea ice product itself against *in situ* measurements, as the overlap between coverage periods is too short. During archive processing we use final sea ice concentration from NSIDC (Cavalieri et al., 1996, updated yearly), rather than the NRT concentration data used in our NRT sea ice calculations. Aside from this, the CryoSat-2 SAR and SARIn mode data are processed identically to the NRT case.

First, we assessed our processing at orbit-scale by calculating point-by-point differences of NRT and archive sea ice freeboards using a single track of CryoSat-2 data from April 2015, for which all geophysical corrections were present in both datasets. The track consisted of 3,968 lead and 5,246 freeboard measurements for the NRT data compared with 3,970 lead and 5,242 freeboard measurements for the archive data. Along this track, NRT and archive freeboards showed excellent agreement, with a mean difference of 0.02 cm (Fig. 2a). We then compared sea ice thickness and volume based on the NRT and archive products, using seven months of data acquired between October 2014 and April 2015, which corresponds to a season of ice growth.. The thickness comparison was done over the 5 km square grid on which NRT data are output. In general, our NRT and archive estimates of sea ice thickness are in excellent agreement, with a mean difference of 0.9 cm (Fig. 2b). NRT and archive estimates of sea ice volume are also in excellent agreement, with an average difference of 175 km³ (Fig. 2c) across the entire Arctic region. The negative freeboard and thickness values apparent in Fig. 2a and Fig. 2b respectively are a consequence of negative freeboard measurements that occur due to random noise in radar echoes from thin ice floes, caused by radar speckle. These freeboards are included in our processing to ensure that the average freeboard, and therefore thickness, is not biased high. Overall, differences between NRT and archive estimates of sea ice thickness and volume fall well within the corresponding estimates of their uncertainties (Tilling et al., 2015).

Our archive estimates of sea ice volume are larger than the NRT product in part as a consequence of using the final sea ice concentration data set, which contains higher values than its NRT counterpart. For example, we recalculated sea ice volume using the NRT sea ice thickness and final sea ice concentration data sets, and the departure from the archive estimate reduced to 100 km³. A contribution to the remaining difference is likely the combined absence of the wet tropospheric, dry tropospheric and inverse barometer corrections in 93.8% of the Baseline-B fast delivery CryoSat-2 data. This is reduced to 0.3% for Baseline-C data. The mean sea ice thickness for both the NRT and archive datasets is ~1.8 m, and there is no bias between them, with or without geophysical corrections applied. When the corrections are missing the NRT and archive thickness values at any given location differ, on average, by just 1.1 cm with a standard deviation of 23.0 cm (Fig. 3a-c). This is reduced to 0.1 cm with a standard deviation of 7.4 cm when the corrections are present (Fig. 3d-f). There is no spatial pattern to these differences. Despite the improvement in performance of Baseline-C NRT data compared with Baseline-B we conclude that the satellite orbits and on-ground processing applied to fast delivery CryoSat-2 data are sufficient to determine accurate measurements of Arctic sea ice thickness and volume for both baselines. The thickness differences between the archive and NRT data products are not significant for either baseline given the estimated uncertainty on thickness and the typical thickness of sea ice floes.

## 3      Results

The spatial distribution of the NRT sea ice thickness data (Fig. 1) for any given time period depends on the nature of the CryoSat-2 orbit over that period. CryoSat-2 has an orbit repeat period of 369 days, which is built up by successive shifts of a 30-day repeat sub-cycle, meaning that uniform coverage of the Arctic Ocean is achieved every 30 days (Wingham et al., 2006). The density of orbit crossovers increases with latitude up to the CryoSat-2 limit of 88°N, and also with the number of days of coverage. We produce Arctic-wide maps of NRT sea ice thickness for the previous 2, 14, and 28 day periods. CryoSat-2 orbit patterns are visible in maps of thickness for the final 2 (e.g. Fig. 1a and Fig. 1d) and 14 (e.g. Fig. 1b and Fig. 1e) days of each month. The orbits are clearer at lower latitudes, below about 80°N. Over 28 days, almost complete coverage across the sea ice pack is achieved. However, there are still small areas of unmapped sea ice, and these typically occur at the ice edge (see Fig. 1). In these unmapped areas the sea ice concentration is above 15%, which we use as the sea ice margin threshold, but below 75%, which is the concentration required for a region to be classed as containing sea ice (see Data and Methods).

To determine the utility of the 5 km grid measurements of NRT sea ice thickness for operational use, we performed a detailed assessment of the spatial and temporal distribution of the data and compared these to the equivalent for archive data. Over the 2, 14 and 28 day time periods for which NRT data are available, we calculated the percentage of sea ice covered by NRT and archive data in 1 degree latitude bands from 60-90°N, for the final 2, 14 and 28 days of each month. This was done for data from October 2014 to April 2015, and averaged over all months (Fig. 4a). We produced the equivalent plot for the mean data separation in each latitude band, where separation is simply the square root of the number of measurements in each band, divided by the sea ice covered area (Fig. 4b). For 28 days data coverage, sea ice at latitudes between 85-88°N is mapped in its entirety by the NRT and archive products and the data separation drops to 5.0 km in each 1 degree latitude band, which is simply the grid separation. For 14 days coverage the CryoSat-2 orbit pattern achieves its maximum coverage for NRT data, of 98%, between 86 and 87°N but achieves 100% coverage for archive data between 86-88 °N. These correspond to a mean data separation of 5.1 km and 5.0 km (the grid separation), respectively. The maximum NRT coverage over 2 days is 91%, between 87 and 88°N, where the mean data separation is 5.2 km. This increases to 99%, between 87 and 88°N for archive data,

with a mean data separation of 5.1 km. For both NRT and archive data the percentage of ice mapped decreases with decreasing latitudes, and the separation between data points increases, although there is some fluctuation in these trends that is likely due to the shift in the CryoSat-2 orbit pattern producing less favourable coverage for a given month. CryoSat-2 does not observe
5    sea ice north of 88°N, so the percentage of ice mapped drops to 0% for 2, 14 and 28 days coverage in the region 88-90°N for both datasets. On average, the NRT sea ice thickness data maps 20, 51 and 66% of the Arctic sea ice north of 60°N every 2, 14 and 28 days respectively. This corresponds to a measurement within 14, 7 and 6 km of each location in the Arctic every 2, 14 and 28 days. For archive data the coverage increases to 23, 57 and 69% every 2, 14 and 28
10   days respectively, which corresponds to a measurement within 13, 7 and 6 km of each location in the Arctic.

The distribution of our NRT sea ice thickness measurements also varies with oceanographic basin and month, and the nature of the monthly variation depends on the region being observed. This is an important consideration for those wishing to use the data in a specific
region of interest, or over the entirety of the sea ice growth season. We calculated the percentage of ice cover mapped by our NRT product for six key oceanographic basins (Fig. 5a), for the final 28 days of each month of the 2014-2015 sea ice growth season (Fig. 5b) and compared this to the percentage of ice cover mapped by our archive data (Fig. 5c). The percentage of the ice cover mapped in the Amerasian and Eurasian basins is high (≥ 76% for
NRT data and ≥ 83% for archive data), with just a small increase over the growth season. Both regions are almost entirely covered in sea ice year-round, which means that the areal fraction of unmapped sea ice at the ice edge (see Fig. 1) is fairly consistent throughout the year. However, this is not the case for regions with more seasonal ice cover, such as the Canadian Archipelago and Northwest Passage, Hudson Bay, and the Beaufort Sea, where NRT and archive coverage
improves throughout the growth season and peaks in February or March. In these regions, as the extent of the sea ice cover increases through winter, the unmapped area at the sea ice edge becomes a decreasing fraction of the ice-covered area, and a greater percentage of the ice cover is mapped. In addition, as the sea ice concentration increases through winter, echoes from sea ice floes becomes less noisy and are more likely to be included in our processing. Coverage in
the Greenland Sea generally improves throughout the growth season, although there is some variation in this pattern due to fluctuations in the width of the unmapped area at the sea ice edge, which could be a consequence of the rapid sea ice transport in this sector. Overall, coverage is lowest for the Greenland Sea, Canadian Archipelago and Northwest Passage, and Hudson Bay. Due to the location of the Greenland Sea, there is also a persistent presence of
unmapped sea ice along its eastern edge. The Canadian Archipelago and Northwest Passage, and Hudson Bay are in close proximity to substantial coastal areas, where it is difficult to construct sea surface height due to the absence of leads in the sea ice pack. Although there is spatial variation in the coverage of our NRT sea ice thickness data, both with latitude (Fig. 4) and oceanographic basin (Fig. 5b), there is no significant spatial variability in the difference between
the NRT and archive data coverage (Fig. 4 and Fig. 5c).

We extended our analysis of NRT data sampling by calculating the percentage of sea ice mapped in all Arctic Ocean basins at the beginning and end of the sea ice growth season (Table 1). For this calculation, we considered the percentage of ice cover mapped in the final 2, 14 and 28 days of each month. In each month the coverage improves with the number of days sampling, in
every basin. The coverage also improves from October to March, for each time period, for all but one basin; the Canadian Archipelago/Northwest Passage experiences a drop in coverage over the growth season, for the 2-day observation period. However, this change is very small, and over short observation periods we would expect some variability in the proportion of ice cover mapped as a consequence of the CryoSat-2 orbital repeat pattern. This becomes more important
in regions such as the Canadian Archipelago, where there is a high fraction of land interspersed with ocean. The Bering Sea, the Sea of Okhotsk, the White Sea, the Baltic Sea and surrounding Gulfs and the Labrador Sea have the smallest proportional ice cover mapped in March 2015.

These are regions of highly seasonal sea ice cover, and by the end of the growth season the unmapped area at the ice edge still constitutes a sizable fraction of the ice–covered area. In addition, they are all southerly basins (below 70°N), which are sampled with reduced spatial density by CryoSat-2. The most extensively sampled areas are in the central Arctic - the Amerasian and Eurasian basins - which experience substantial year-round sea ice cover and are at high latitudes. We conclude that the location, seasonality, and dynamic nature of any sea ice region are important considerations when assessing the reliability of the NRT Arctic sea ice thickness product.

## 4      Discussion and Conclusions

Our CryoSat-2 NRT sea ice thickness dataset will benefit Arctic sea ice projections, because it can be used to constrain physical models that investigate the sensitivity of the region to climate change (Day et al., 2014) in a timely manner. It will also assist Arctic operations that rely on accurate and timely information on sea ice thickness, such as natural resource exploration (Galley et al., 2013), and shipping for cargo (Liu and Kronbak, 2010) and tourism (Stewart et al., 2007). A previous study (Rinne and Similä, 2016) has highlighted the potential value of fast delivery CryoSat-2 data for the classification of sea ice into discrete stages of its development – thin (<70 cm) and thick (>70 cm) FYI and MYI – in the Kara Sea. Our product extends this analysis to provide continuous measurements of sea ice thickness across the entire northern hemisphere, complementing established records of sea ice concentration (Cavalieri et al., 1996, updated yearly, Maslanik and Stroeve, 1999, updated daily) upon which annual assessments (Stroeve et al., 2005) and forecasts (Posey et al., 2011) of Arctic conditions are based. Timely availability of sea ice concentration estimates (Maslanik and Stroeve, 1999, updated daily) and sea ice type classifications (http://osisaf.met.no/p/ice/#type) are crucial for the rapid computation of our NRT sea ice thickness measurements. The NSIDC sea ice concentration and OSISAF sea ice type data are available to us by 01:00 UTC two days after, and 01:00 UTC the day after measurement, respectively. The fast delivery CryoSat-2 data are typically available 36 hours after acquisition from the satellite, but can vary from 1-3 days, so we run our sea ice processor at a latency of three days to ensure sufficient data is available. Processing one day of data for the northern hemisphere takes six hours, on average. A more rapidly delivered product would require the CryoSat-2 data to be consistently available within 36 hours, and sea ice concentration data to become available sooner, or that older concentration measurements were used as an approximation.

By using a new fast delivery CryoSat-2 dataset we are able to produce estimates of sea ice thickness across the northern hemisphere three days after acquisition from the satellite. This marks the beginning of a new phase for the CryoSat-2 mission, in which its primary data can be used for operational purposes. The NRT estimates are of comparable accuracy to those produced using the final release CryoSat-2 data, with a mean difference of 0.9 cm between NRT and archive estimates of sea ice thickness. The NRT and archive thickness differences, although small, vary temporally. The differences are reduced when all geophysical corrections are present in the fast delivery CryoSat-2 data, which is the case in 99.7% of the data since March 26th 2015, when the ESA on-ground processing chain switched from Baseline-B to Baseline-C. There is no spatial variability in the differences between our NRT and archive data products. For the period from October 2014 to April 2015, the NRT dataset covers an average of 20, 51 and 66% of the Arctic sea ice north of 60°N every 2, 14 and 28 days respectively. This is equivalent to a measurement within 14, 7 and 6 km of each location in the Arctic every 2, 14 and 28 days. However, there are temporal and spatial variations in the data coverage. The time of year, location, and dynamic nature of any region of interest must be considered when assessing the reliability of the data. The next major step in the advancement of the data is to develop improved estimates of snow loading on Arctic sea ice. We also intend to investigate the impact of different gridding methods, including the application of a distance weighting, on our gridded NRT sea ice thickness product. Our sea ice thickness and volume error budget could be further

constrained with improved knowledge on uncertainties in snow loading and sea ice density, and also by accounting uncertainties in the propagation speed of the radar signals through the snow pack. We encourage users to utilise the data for model assessments and to constraint the physics of sea ice within models that form the basis of future climate projections.

## Author contribution

R. L. Tilling and A. Ridout developed and analysed the satellite observations. A. Shepherd supervised the work. R. L. Tilling, A. Ridout and A. Shepherd wrote the paper. All authors commented on the text.

## Acknowledgements

Our NRT sea ice thickness data are publically available at http://www.cpom.ucl.ac.uk/csopr/seaice.html. We wish to thank those who provide the timely ancillary data that we require to deliver a NRT product: ESA, for the fast delivery CryoSat-2 Level 1B radar altimeter data (available via ftp at ftp://science-pds.cryosat.esa.int); OSI SAF, for their sea ice type maps (http://osisaf.met.no/p/ice/#type); and NSIDC, for NRT DMSP SSMIS Daily Polar Gridded Sea Ice Concentrations (available via ftp at ftp://sidads.colorado.edu/pub/DATASETS/nsidc0081_nrt_nasateam_seaice). This work was funded by the UK Natural Environment Research Council, with support from the UK National Centre for Earth Observation.

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

Table 1: Variations in the sampling of CryoSat-2 near real time (NRT) sea ice thickness products in 17 Arctic Ocean basins. Regions 1-10 encompass all October sea ice, and regions 1-16 encompass all March sea ice. Region 17 is a sub-region of region 1 (Figure 5a).

| | Data Coverage (% of ice cover mapped) | | | | | |
|---|---|---|---|---|---|---|
| | **2 days** | | **14 days** | | **28 days** | |
| | **Oct 2014** | **Mar 2015** | **Oct 2014** | **Mar 2015** | **Oct 2014** | **Mar 2015** |
| Amerasian Basin (1) | 33 | 38 | 78 | 82 | 92 | 98 |
| Eurasian Basin (2) | 24 | 44 | 58 | 73 | 76 | 88 |
| Canadian Archipelago & Northwest Passage (3) | 9 | 7 | 31 | 37 | 39 | 53 |
| Hudson Bay (4) | 0 | 6 | 0 | 48 | 0 | 71 |
| Baffin Bay (5) | 0 | 15 | 0 | 56 | 0 | 81 |
| Greenland Sea (6) | 8 | 13 | 31 | 50 | 49 | 63 |
| Iceland Sea (7) | 0 | 16 | 0 | 44 | 0 | 57 |
| Barents Sea (8) | 0 | 9 | 17 | 32 | 18 | 47 |
| Kara Sea (9) | 2 | 17 | 15 | 46 | 16 | 58 |
| Siberian Shelf Seas (10) | 11 | 20 | 38 | 60 | 49 | 85 |
| Bering Sea (11) | n/a | 3 | n/a | 35 | n/a | 40 |
| Sea of Okhotsk (12) | n/a | 0 | n/a | 21 | n/a | 33 |
| White Sea (13) | n/a | 0 | n/a | 6 | n/a | 6 |
| Baltic Sea & surrounding Gulfs (14) | n/a | 0 | n/a | 0 | n/a | 0 |
| Labrador Sea (15) | n/a | 1 | n/a | 13 | n/a | 19 |
| Gulf of St Laurence & Nova Scotia Peninsula (16) | n/a | n/a | n/a | n/a | n/a | n/a |
| Beaufort Sea (17) | 17 | 20 | 59 | 83 | 69 | 95 |

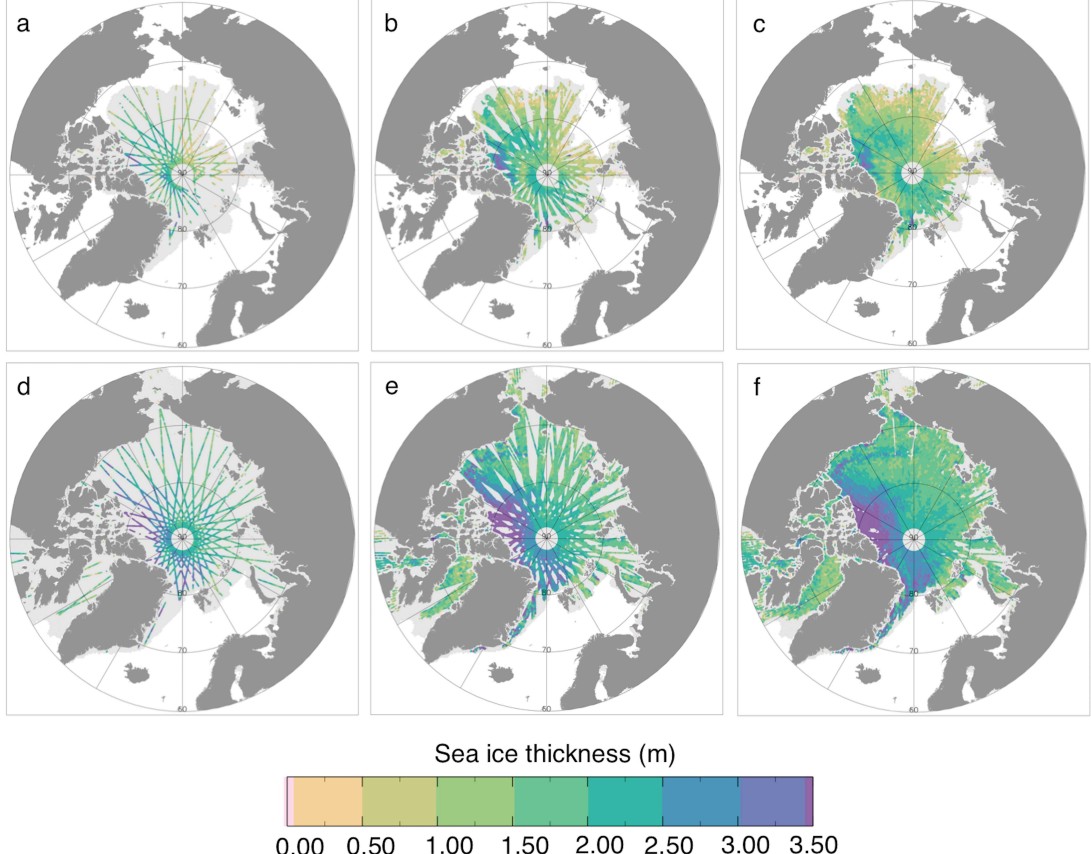

Figure 1: Near real time (NRT) Arctic sea ice thickness estimates from CryoSat-2. (a)-(c) Thickness estimates for the final 2, 14 and 28 days in October 2014, respectively. (d)-(f) Thickness estimates for the final 2, 14 and 28 days in March 2015, respectively. NRT sea ice thickness data are output Arctic-wide on a 5 km square grid. All thickness measurements within a 25 km radius of the centre of the grid are averaged, with all points receiving equal weight. The sea ice extent mask is shaded in light grey, and highlights unmapped areas of the sea ice.

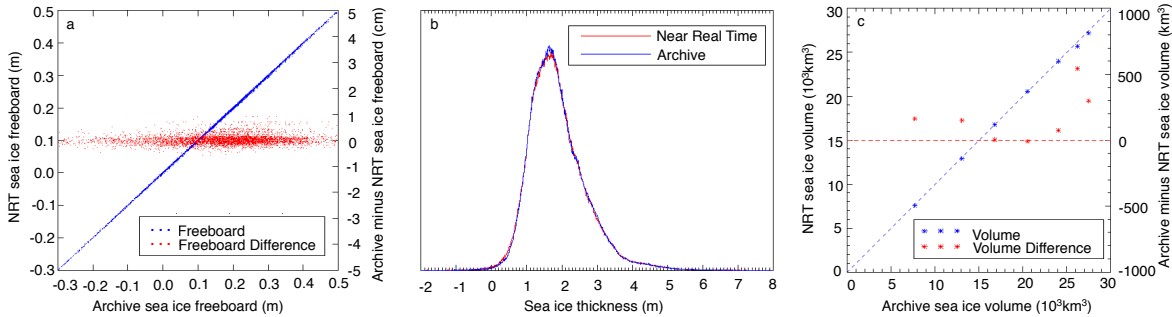

Figure 2: Comparison of near real time (NRT) and archive estimates of Arctic sea ice freeboard, thickness, and volume, from CryoSat-2. (a) Crossplot of point-by-point sea ice freeboard for an Arctic pass in April 2015. Also shown is the difference (archive minus NRT) in sea ice freeboard between the datasets. (b) Normalised distribution of NRT and archive thickness estimates over the period October 2014-April 2015, for all grid cells where measurements are available for both datasets. (c) Crossplot of sea ice volume for October 2014-April 2015. Also shown is the difference (archive minus NRT) in sea ice volume between the datasets.

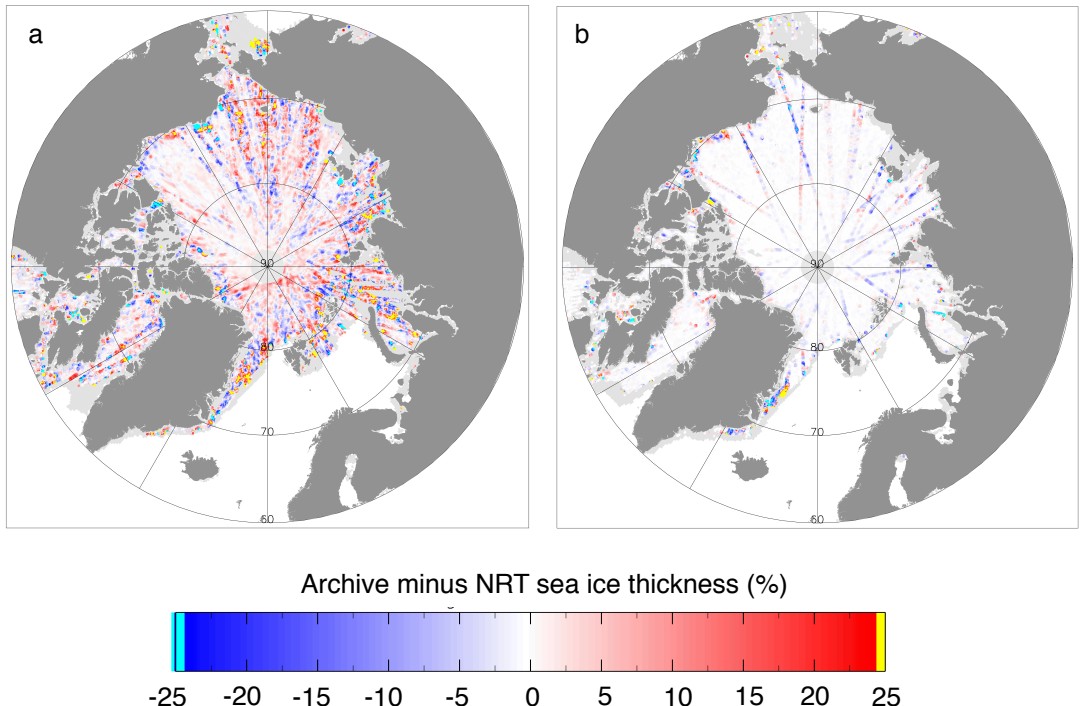

Figure 3: The impact of geophysical corrections on near real time (NRT) Arctic sea ice thickness estimates from CryoSat-2. (a) Percentage change in archive minus NRT thickness estimates for the final 28 days of March 2015. In March 2015 the wet tropospheric, dry tropospheric and inverse barometer corrections were missing in 80% of cases. (d) Percentage change in archive minus NRT thickness estimates for the final 28 days of April 2015. In April 2015 the wet tropospheric, dry tropospheric and inverse barometer corrections were missing in 0% of cases.

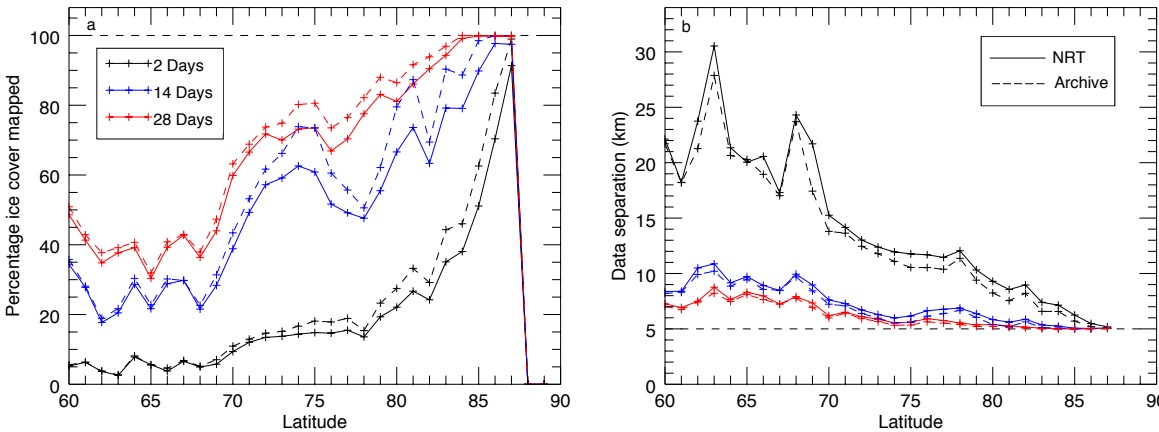

Figure 4: Spatial and temporal sampling of the Centre for Polar Observation and Modelling (CPOM) near
real time (NRT) and archive Arctic sea ice thickness products, north of 60°N. (a) Plot showing the
percentage of sea ice cover mapped in 1° latitude bands, averaged over each month from October 2014-
April 2015. Data are plotted for the final 28, 14, and 2 days of all months. Solid lines = NRT data, dashed
lines = archive data. (b) Plot showing the mean separation between NRT measurement points in 1°
latitude bands, averaged over each month from October 2014-April 2015. Data are plotted for the final 28,
14, and 2 days of all months. Solid lines = NRT data, dashed lines = archive data.

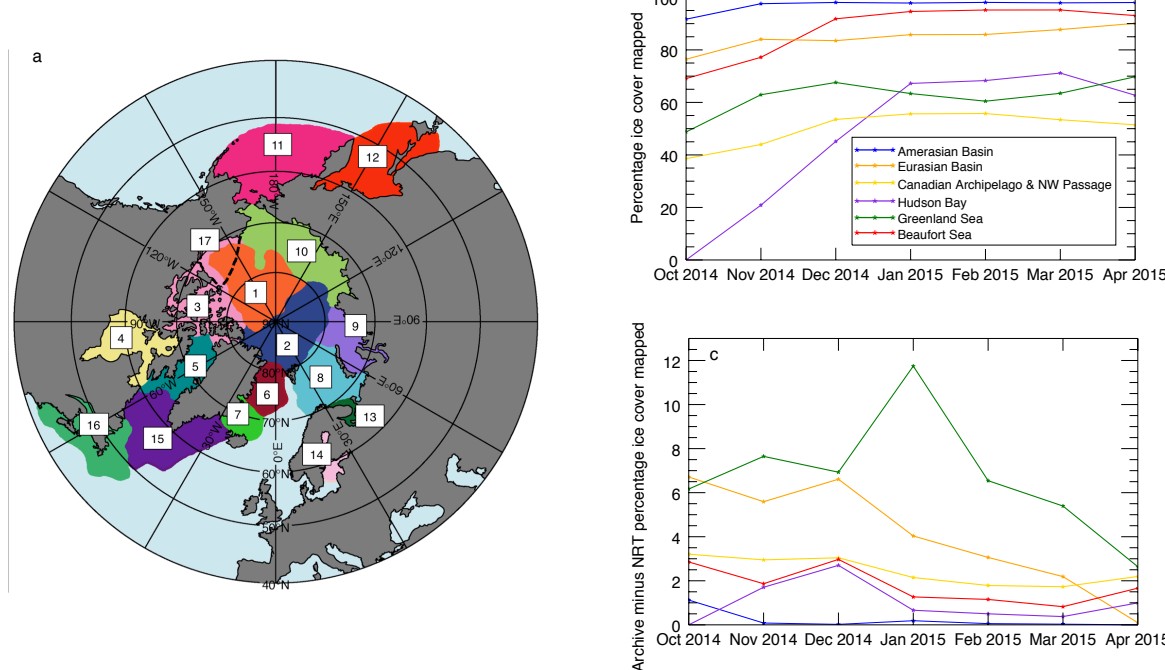

Figure 5: Regional and temporal sampling of the Centre for Polar Observation and Modelling (CPOM) near real time (NRT) and archive Arctic sea ice thickness products. (a) Arctic Ocean regions selected for analysis. The regions are the Amerasian Basin (1), Eurasian Basin (2), Canadian Archipelago and Northwest Passage (3), Hudson Bay & Foxe Bay (4), Baffin Bay (5), Greenland Sea (6), Iceland Sea (7), Barents Sea (8), Kara Sea (9), Siberian Shelf Seas (10), Bering Sea (11), Sea of Okhotsk (12), White Sea (13), Baltic Sea & surrounding Gulfs (14), Labrador Sea (15), the Gulf of St Lawrence & Nova Scotia Peninsula (16), and the Beaufort Sea (17). Regions 1-10 encompass all autumn sea ice, and regions 1-16 encompass all spring sea ice. Region 17 is a sub-region of region 1 and 3. (b) Plot showing the percentage of sea ice cover mapped by the NRT product in each month, for six key oceanographic basins. (c) Plot showing the difference (archive – NRT) in percentage ice cover mapped.