# Peer review of "Near Real Time Arctic sea ice thickness and volume from CryoSat-2"

_The Cryosphere, 2016_

## Referee Comment (RC1) · Anonymous Referee #1 · 28 Feb 2016

General comments:

This study considers the use of CryoSat-2 Near-Real-Time (NRT) data from ESA, for calculating sea ice thickness and volume. The authors present a quality assessment by comparing the NRT product with the ice thickness and volume derived from the archived level1b data from ESA. They analyse this data for one sea ice growth season from October 2014 to April 2015 and conclude that NRT ice thickness is of comparable accuracy to the product, derived from archived data. The authors pronounce the benefit of NRT thickness data for climate models and industry, e.g. shipping and tourism.

Indeed, the availability of NRT CryoSat-2 thickness is a great achievement. However, the methods to derive sea-ice thickness (and volume) are the same as described in Tilling et. al. (2015). Many parts of the methods are already discussed in the Nature

paper. The novel part is given by statistical analysis of data coverage and differences in thickness and volume of the NRT product on a basin scale.

In general, my concern is that this manuscript lacks of an in-depth analysis. The focus of this paper is set on the comparison and difference between the NRT and the final released product. But more elaboration of these differences is needed. There are substantial questions that I think should be addressed, going along with further analysis:

1. The volume comparison in Figure 2 reveals higher values for the final release product. You state that this is mostly because of the the use of different ice concentrations, but also due to the absence of orbits in the NRT level1b data. Nevertheless, Figure 1, 3, 4 and Table 1 only show statistics with respect to the NRT product. Can you include the same statistics for the final release product (as in Figure 3 and Table 1 for the NRT product) and also the different ice concentrations, you used? I think this is needed in order to proof your statement above and to turn out the differences.

2. Although many readers are interested only in the final thickness product, comparing only the thickness histograms of both products, is not enough from my point of view. I suggest to show freeboard (and thickness) maps of difference between the NRT and the archive product in autumn and spring. This would give further information about the spatial distribution of differences between both products.

3. The CS-2 data processing starts with the NRT level1b data and the processing of each orbit segment. Therefore I would suggest also to consider differences on the orbit-scale, like the comparison of freeboard along track between both products or even just the comparison between the ellipsoidal elevations (after retracking). And what about the detected leads? Is it the same for both products?

Detailed comments:

P2 L38: The oil and gas sector requires sea ice information for feasibility studies. Why

is the reduction of plans for exploration and drilling a consequence? I think it needs one more sentence to explain this.

P2 L28-30: So you use NRT SAR and SIN, right? Is there a difference between handling both modes in the NRT product. Or to be more specific, are the differences between NRT SAR and archive SAR the same as between NRT SIN and archive SIN? Would it make sense to separate between the modes in this study?

P4 L5: Can you be more specific: Which geophysical corrections are missing in the fast delivery data? What does 'often' mean in this statement?

P4 L15-19: How do you justify using the Warren climatology in regions where W99 is not based on measurements, for example in the Baffin Bay. W99 is a 2d fit and therefore it is not constraint in such areas and can produce substantial biases which are not considered in the uncertainty estimates. In some ares like Barents Sea in November, it can even cause negative snow depths.

P4 L27-29: Why do you use the same weighting for all points? If you project on a 5 km grid, but using a 25 km radius for averaging, this means that the grid cell covers only 1% of the area which goes into the average (5x5 km = 25 km^2, pi x (25km)^2 = 1963 km^2)? Is that right? But then the grid cell is hardly representative for the thickness at this location. What is the circular operator doing? Would it make sense to apply a distance weighting?

P4 L33-34: How is the gap filled at the pole?

P6 L1: ... absence 'o'f ...

Figure 3: Can you add the data for the final release product? I think it would be helpful to understand the differences in coverage between both products.

Figure 4b: Can you add the data coverage of the final release product (see previous comment)?

[Figure]

---

## Short Comment (SC1) · 4 Mar 2016

Hi, My name is Charlie Rodriguez and I am a GIS/Analyst. I was wondering if you were going to create a time domain analysis of the Ice sheets? I used data from the 1969 Nimbus imagery for my senior project to show the waxing and waning of the ice for a nine month period. Will you be conducting any type of mapping to show deteriorating of the arctic sea ice thickness? I am keen on being able to see this type of data being presented to the public as I have to face "Global warming deniers" every day and your data could help me with my arguments. Thanks so much. Charlie

---

## Referee Comment (RC2) · Anonymous Referee #2 · 18 Mar 2016

This article presents near-real time Arctic sea ice thickness and volume estimates from CryoSat-2 data. The availability of a NRT data product is a great achievement which could have wide benefit. Much of the article describes the differences between the NRT product and regular product, and uncertainties for the NRT product are also determined. The paper also discusses sea ice volume determination, this type of product is much more of interest for the comparison of current conditions to long-term climate records rather than a tool for shipping, oil, or other resource extraction uses. I believe a product such as this, and particularly the associated uncertainties, require a much more detailed treatment than what has presently been done. Ingestion and comparison to models also requires that the data biases and uncertainties be well known and described. But as detailed below, I believe that the uncertainties in the data are larger than were presented in the paper due oversimplification of errors as well as the

possible exclusion of key uncertainty factors.

In several areas of the text the mathematical operations performed on the data need to be explicitly written out as otherwise it is unclear exactly how some of the calculations were done. One example on this is that it is unclear whether a correction for the slower speed of light in snow has been applied to the calculation of freeboard. It is stated in Tilling et al., 2015 "A correction is applied to each freeboard measurement to account for the attenuation of the radar pulse as it passes through any snow cover on sea ice, where snow depth is based on a climatology." But this sentence is confusing as it could also apply to attenuation of energy through the snow, which in itself would not necessarily impact the freeboard determination. If this factor is applied, and whether it was applied in the determination of sea ice thickness and volume uncertainty, is not clear in the text.

It is also unclear how freeboard retrieval errors would propagate into the uncertainty calculations. Tilling et al., 2015 state that an interpolation is done between ocean surface elevation measurements to determine freeboard. The interpolation procedure was not explicitly stated but needs to be done so here. Any such interpolation would change the correlation length of the errors in the assessment and needs to be considered.

Further detailed comments are outlined below:

P2L25: The need for model ingestion is mentioned. But it should be considered that many models which ingest data have trouble with gridded mean sea ice thickness data and prefer to work with swath level data because sea ice thickness in modern models is represented as a distribution rather than a mean value. It would be more useful to provide the point to point measurements of freeboard (the actual measurement made by CryoSat-2) which could be more easily ingested in a model.

P4: The mathematical expression for determination of sea ice thickness error needs to be written out. Was the uncertainty due to the lower speed of light in snow considered in the error estimates?

P4L27: The mathematical expression for the circular operator needs to be written out as it is unclear how this was applied to the data.

P3L19: The reference to Kwok et al., 2009 is confusing here as the paper does not describe the use of CryoSat-2 data.

P4L5: Which geophysical corrections are often missing in the data? They should be listed.

P4L16-17: How is snow from the Warren climatology applied beyond areas of the central Arctic? The reasons for this were mentioned clearly in the other review. I think this is a critical part of the manuscript as this could have a large impact on first year ice areas outside of the central Arctic basin.

P4L17-18: The specific densities for sea ice and water need to be written out.

P4L26: If a 1 km grid can be provided, why not also provide the swath level freeboard data which is of similar resolution?

P4L37: Given the extrapolations of the Warren climatology outside of the central Arctic, as well as the modified version over first year ice, I would question these snow depth uncertainty estimates as they have been quite modified from their original source.

P4L42-44: The statement that the large number of freeboard measurements negates the uncertainty rests on the assumption that the errors are uncorrelated in space and time. This seems highly unlikely given that the retrieval method does not account for factors such as changing snow conditions as shown by Ricker et al., 2015.

Ricker, R., Hendricks, S., Perovich, D. K., Helm, V., & Gerdes, R. (2015). Impact of snow accumulation on CryoSat‐2 range retrievals over Arctic sea ice: An observational approach with buoy data. Geophysical Research Letters, 42(11), 4447-4455.

P5L1-7: The method for determining volume uncertainties is unclear and should be written out mathematically to fully describe the procedure. Also, over what range is

each parameter adjusted to calculate the rate of change?

P5 second paragraph: I think this estimate of error is a gross simplification of the uncertainties and is not accurate. For the snow depth term, it was already acknowledged that there are large differences over first year and multi-year ice which are unrelated to synoptic scale meteorology but is rather related to the timing of snow fall events and ice freeze-up. Sea ice density would also similarly be unrelated to synoptic scale meteorology particularly as the values used in the study are based on first year and multi-year ice types. I would therefore not consider the 2000 km decorrelation length to be accurate. Have you looked at other data to determine the decorrelation length for these parameters?

The last sentence in this paragraph is not accurate as there is likely residual error in the sea surface height estimate since there is a need to interpolate over data gaps due to the varying number of lead points available. The interpolation procedure needs to be written out so that the correlation length of errors in the sea ice thickness can be better understood and taken into account.

Figure 2a: There appear to be negative ice thickness values in the distribution, I'm guessing this is due to uncertainties in the freeboard retrieval but some explanation on this is in order.

A map of the differences with the final data compared to the NRT also needs to be shown. This will reveal whether regional differences are present.

---

## Author Comment (AC1) · 27 Apr 2016

**Response to Referee 1's comoments**

Below we summarise the comments of Referee 1, along with our responses and actions:

| # | Comment (verbatim) | Response | Action |
|---|---|---|---|
| **R1.1** | "In general, my concern is that this manuscript lacks of an in-depth analysis. The focus of this paper is set on the comparison and difference between the NRT and the final released product. But more elaboration of these differences is needed." | We agree that the paper would benefit from a more in-depth analysis of the differences between our NRT and archive data products. Please see our response to **R1.2** and **R1.4** for specific examples. | We have expanded our comparison of our NRT and archive data products. Please see action for **R1.2** and **R1.4** for specific examples. |
| **R1.2** | "The volume comparison in Figure 2 reveals higher values for the final release product. You state that this is mostly because of the use of different ice concentrations, but also due to the absence of orbits in the NRT level1b data. Nevertheless, Figure 1, 3, 4 and Table 1 only show statistics with respect to the NRT product. Can you include the same statistics for the final release product (as in Figure 3 and Table 1 for the NRT product) and also the different ice concentrations, you used? I think this is needed in order to proof your statement above and to turn out the differences." | We agree that readers may desire more information on differences between NRT and archive sea ice thickness products. After further inspection, we find that it is the absence of certain geophysical corrections (wet tropospheric, dry tropospheric and inverse barometer), rather than orbits, that drive the remaining differences in sea ice thickness and volume. This can be shown by plotting the spatial variability of these differences for two different months: one with corrections absent and one with corrections present. | We have included a new figure (Figure 3), which consists of 2 maps, detailing the spatial differences between NRT and archive sea ice thickness for data absent and present geophysical corrections. The explanatory text for this figure (Data and Methods final paragraph, final few sentences) reads:

*"The remaining difference is likely due to the combined absence of the wet tropospheric, dry tropospheric and inverse barometer corrections in 93.8% of the Baseline-B fast delivery CryoSat-2 data. This is reduced to 0.3% for Baseline-C data. The mean sea ice thickness for both the NRT and archive datasets is ~1.8 m, and there is no bias between them, with or without geophysical corrections applied. When the corrections are missing the NRT and archive thickness values at any given location differ, on average, by 1.1 cm with a standard deviation of 23.0 cm (Figure 3a). This is reduced to 0.1 cm with a standard deviation of 7.4 cm when the corrections are present (Figure 3b). There is no spatial* |

| | | | *pattern to these differences. Despite the improvement in performance of Baseline-C NRT data compared with Baseline-B we conclude that the satellite orbits and on-ground processing applied to fast delivery CryoSat-2 data are sufficient to determine accurate measurements of Arctic sea ice thickness and volume for both baselines. The thickness differences between the archive and NRT data products are not significant for either baseline given the estimated uncertainty on thickness and the typical thickness of sea ice floes."*

We have also added archive data to figures 3 and 4b (figures 4 and 5b in updated version), with discussion in the relevant places. Please see action to **R1.12** and **R1.13** for more details.

We have also included a description of the spatial and temporal differences between NRT and archive sea ice thickness data in our Discussion and Conclusions section, second paragraph. This reads:

*"The NRT and archive thickness differences, although small, vary temporally. The differences are reduced when all geophysical corrections are present in the fast delivery CryoSat-2 data, which is the case in 99.7% of the data since March 26th 2015, when the ESA on-ground processing chain switched from Baseline-B to Baseline-C. There is no spatial variability in the differences between our NRT and archive data products."* |
|---|---|---|---|
| **R1.3** | "Although many readers are interested only in the final thickness product, comparing only the thickness histograms of both products, is not enough from my point of view. I suggest to show freeboard (and thickness) maps of difference | Agreed. Please see response to **R1.2** | Please see action to **R1.2** |

| | | | |
|---|---|---|---|
| | between the NRT and the archive product in autumn and spring. This would give further information about the spatial distribution of differences between both products." | | |
| **R1.4** | "The CS-2 data processing starts with the NRT level1b data and the processing of each orbit segment. Therefore I would suggest also to consider differences on the orbit-scale, like the comparison of freeboard along track between both products or even just the comparison between the ellipsoidal elevations (after retracting). And what about the detected leads? Is it the same for both products?" | We agree that there is likely to be interest in the accuracy of our NRT data on an orbit-scale, and so we have included further illustrations and analysis of this in our revised paper. We feel that an along-track comparison of sea ice freeboard is sufficient, as the differences in sea surface heights at the leads will form part of the small differences seen in freeboard.

If the referee is asking whether there is a difference in the number of leads detected in the NRT product compared to the archive then we can include this in our revision, but it is not clear from the question. | We have added an additional panel to Figure 2. Figure 2a now shows the point-by-point freeboard differences for our archive and NRT data products for an individual Arctic pass. This has been described in the final Data and Methods paragraph:

*"Firstly we assessed our orbit-scale processing by calculating point-by-point differences of NRT and archive sea ice freeboard using one track of CryoSat-2 data from April 2015, for which all geophysical corrections were present in the NRT and archive data. These showed excellent agreement, with an average difference of 0.1 cm (Fig. 2a)."* |
| **R1.5** | **P2 L38:** "The oil and gas sector requires sea ice information for feasibility studies. Why is the reduction of plans for exploration and drilling a consequence? I think it needs one more sentence to explain this." | We agree that this sentence would benefit from further justification, and so we have done this in our revised paper. | We have added an extra sentence that reads:

*"Without these studies companies cannot be sure that their infrastructure is suitably robust for the Arctic environment, such as when the Shell oil rig Kulluk ran aground in January 2013."* |
| **R1.6** | **P2 L28-30**: "So you use NRT SAR and SIN, right? Is there a difference between handling both modes in the NRT product. Or to be more specific, are the differences between NRT SAR and archive SAR the same as between NRT SIN and archive SIN? Would it make sense to separate | We agree that it is not clear in the paper which data modes we use, how we use them, and whether this differs for NRT and archive thickness processing. We have done this in our revised paper. | We have added an explanation of the way in which we process SAR and SARIn data for NRT situations. The first Data and Methods paragraph, first five sentences, now read:

*"We use fast delivery radar altimeter measurements from the ESA CryoSat-2 satellite [Wingham et al., 2006] synthetic aperture radar (SAR) and SAR interferometric (SARIn) mode data products to produce NRT estimates of Northern* |

| | | | *Hemisphere (latitudes above 40° N) sea ice thickness and volume. The data are Level 1b, and consist of an echo for each point along the ground track of the satellite. For Arctic sea ice processing we assume that the ice surface is relatively flat and that slope variations are minimal [Rapley et al., 1983], so are concerned principally with power returns from nadir. Therefore SARIn mode waveforms are cropped to include only the central 128 range bins. This allows for identical processing of SAR and SARIn mode data as both now have 128 bins in their waveform data."* |
|---|---|---|---|
| | | | We have also clarified that our processing of SAR and SARIn data is the same for NRT and archive cases. There is now a sentence in the final paragraph of Data and Methods that reads: |
| | | | *"Aside from this, the CryoSat-2 SAR and SARIn mode data are processed identically to the NRT case."* |
| R1.7 | **P4 L5:** "Can you be more specific: Which geophysical corrections are missing in the fast delivery data? What does 'often' mean in this statement?" | We agree that it would be helpful to be specific about which geophysical corrections are missing, and so we have done this in our revised paper. | The sentence in question has been expanded to read: *"In the fast delivery data the wet tropospheric, dry tropospheric and inverse barometer corrections are missing in 93.8% of cases for Baseline-B data, but only 0.3% of cases for Baseline-C data. In these cases, all three of the corrections are missing. "* We have also moved the sentence further up in the paragraph as we feel it makes more sense to include it immediately after the baseline processing is introduced. |
| R1.8 | **P4 L15-19:** "How do you justify using the Warren climatology in regions where W99 is not based on measurements, for example in the Baffin Bay. W99 is a 2d fit and therefore it is not constraint in such areas and can produce substantial biases | We realise now that our treatment of the Warren climatology and our justification of its use are not clearly explained.  We share the referee's concerns regarding the Warren climatology, especially in regions where it is not constrained by *in situ* measurements. Hence we use the mean climatology values of snow depth and density from a | A sentence has been added to summarise our treatment of the Warren climatology. It reads: *"To obtain snow depth and density we average the values from a climatology (Warren et al. 1999) that fall within the ICESat domain, where the climatology is constrained by in situ* |

| | | | |
|---|---|---|---|
| | which are not considered in the uncertainty estimates. In some areas like Barents Sea in November, it can even cause negative snow depths." | fixed central Arctic domain (where snow parameters are constrained) in all freeboard to thickness conversions, no matter where they are located. There are known differences between the climatology and the current snow depth on younger Arctic sea ice (Kurtz *et al.* 2011; Webster *et al.* 2014) so we halve the snow depth on FYI to account for reduced snow accumulation. Although this approach cannot capture all of the known variability, it removes the possibility of errors being introduced through extrapolation. This detail is now included in our revised paper. | *measurements."*

The ICESat domain itself is defined earlier in the paper.

Should the reader require further information, the second paragraph in the Data and Methods section, first sentence, now reads:

*"The processing steps for fast delivery CryoSat-2 data are identical to those used for the final delivery data, and are described in Tilling et al. (2015)."* |
| R1.9 | **P4 L27-29:** "Why do you use the same weighting for all points? If you project on a 5 km grid, but using a 25 km radius for averaging, this means that the grid cell covers only 1% of the area which goes into the average (5x5 km = 25 km^2, pi x (25km)^2 = 1963 km^2)? Is that right? But then the grid cell is hardly representative for the thickness at this location. What is the circular operator doing? Would it make sense to apply a distance weighting?" | We agree that employing a distance weighting when computing our gridded thickness product may potentially be of benefit (it also may not). However, the aim of this study is not to alter our current processing method. Rather, our aim is to apply our existing method to fast delivery CryoSat-2 data and compare the results to calculations based on archive data, and to do this requires that our processing to remain the same. The effect of gridding methods on gridded sea ice thickness could form the basis of another study. | There is now a sentence in the final Discussion and Conclusions paragraph that reads:

*"We will also investigate the impact of different gridding methods, including the application of a distance weighting, on our gridded NRT sea ice thickness product."* |
| R1.10 | **P4 L33-34:** "How is the gap filled at the pole?" | We realise that our approach for filling the polar gap in volume calculation was not explained. Note that this procedure only applies to the volume calculation in the comparison with archive results, it is not required for the thickness products. | Our sea ice volume method description now includes a sentence that reads:

*"Empty thickness grid cells within the sea ice extent mask, including those north of 88°N, are filled by nearest neighbour interpolation with a maximum search radius of 300 km."* |
| R1.11 | **P6 L1:** "... absence 'o'f ..." | Agreed | Changed to "of" |

| R1.12 | **Figure 3:** "Can you add the data for the final release product? I think it would be helpful to understand the differences in coverage between both products." | We agree that this would be helpful, as would a description of the differences. Both are added to the revised paper. | The final data are now included in the figure (now Figure 4). The second Results paragraph, first sentence, now reads:

*"To determine the utility of the 5 km grid measurements of NRT sea ice thickness for operational use, we performed a detailed assessment of the spatial and temporal distribution of the data and compared these to the equivalent for archive data."*

The paragraph then discusses these comparisons. |
|---|---|---|---|
| R1.13 | **Figure 4b:** "Can you add the data coverage of the final release product (see previous comment)?" | We agree that this would also be helpful, as would a description of the differences. Again, both are added to the revised paper. | The final data are now included in the figure (now Figure 5b). The third Results paragraph, second sentence, now reads:

*"We calculated the percentage of ice cover mapped by our NRT product for six key oceanographic basins (Fig. 5a), for the final 28 days of each month of the 2014-2015 sea ice growth season and compared this to the percentage of ice cover mapped by our archive data (Fig. 5b)."*

The paragraph then discusses these comparisons.

The third results paragraph summarises the new contents of figures 4 and 5b, saying:

*"Although there is spatial variation in the coverage of our NRT sea ice thickness data, both with latitude (Fig. 4) and oceanographic basin (Fig. 5b), there is no significant spatial variability in the difference between the NRT and archive data coverage (Fig. 4 and Fig. 5c)."* |

**References**

Rapley, C. G., et al. (1983), *A study of satellite radar atlimeter operation over ice-covered surfaces; EAS contract report no. 5182/82/f/CG(SC)*, ESA Scientific and Techincal Publication Branch ESTEC Noordwijk, Holland.

Tilling, R. L., A. Ridout, A. Shepherd, and D. J. Wingham (2015), Increased Arctic sea ice volume after anomalously low melting in 2013, *Nature Geoscience*, *8*, 643-646.

Wingham, D. J., et al. (2006), CryoSat: A mission to determine the fluctuations in Earth's land and marine ice fields, in *Natural Hazards and Oceanographic Processes from Satellite Data*, edited by R. P. Singh and M. A. Shea, pp. 841-871.

---

## Author Comment (AC2) · 27 Apr 2016

Hi Charlie. Thank you for your interest in our data. We are able to produce data for seven months of the year, over the Arctic sea ice growth season. Throughout this period the dominant signal in sea ice thickness changes will be due to seasonal variations. CryoSat-2 was launched in 2010 so our time series is currently not long enough to produce data that show long-term interannual changes in the sea ice thickness, or to comment on trends. This is why we need the CryoSat-2 mission to continue or to be followed by similar missions, so that we can start to produce the data that you require, and that will be of great interest to the community.

---

## Author Comment (AC3) · 27 Apr 2016

**Response to Referee 2's comments**

Below we summarise the comments of Referee 2, along with our responses and actions:

| # | Comment (verbatim) | Response | Action |
|---|---|---|---|
| **R2.1** | "I believe a product such as this, and particularly the associated uncertainties, require a much more detailed treatment than what has presently been done."

"I believe that the uncertainties in the data are larger than were presented in the paper due oversimplification of errors as well as the possible exclusion of key uncertainty factors." | There are insufficient observations to fully characterise (i.e. correct for) certain sources of variability in the retrieval of sea ice thickness and volume. Because of this, our estimates of sea ice thickness and volume are in error. Examples include temporal variations in the microwave scattering horizon, spatial variations in snow loading, and temporal variations in the concentration and extent of sea ice. None of these signals have been adequately sampled using independent measurements, and so we cannot be sure of their variance. To account for this, we introduce uncertainties in the key factors of our retrieval based on information present within the published literature. In the case of our archive product, this includes uncertainties in snow depth, snow density, ice density, sea ice concentration, sea ice extent, and sea ice freeboard (which decorrelates rapidly in space)(Tilling et al., 2015). We do not include an uncertainty associated with temporal variations in the microwave scattering horizon (i.e. the difference between the radar and ice freeboard), because these have been shown to rapidly decorrelate with time and to preferentially affect waveform retrackers designed to locate the ice surface (Ricker et al., 2015), which we do not employ. Our error model leads to uncertainties in Arctic-wide sea ice volume of around 15 %, and in sea ice thickness of around 25% at the 25 km scale of our grid. The latter are comparable to the spread of differences between our archive product and independent | We have expanded our error budget to include the contribution of sea ice freeboard uncertainty due to a.) sea surface height uncertainty and b.) floe height measurement uncertainty (due to radar speckle and random noise in the retracking step). Please see our response and action to **R2.3** and **R2.17**.

We have added the treatment of a.) and b.) to the description of our error analysis, and introduce these by stating that

*"The construction of our error budget is described in Tilling et al. [2015], but we now expand on this by considering the contribution of uncertainty in sea ice freeboard in more detail."*

We have been explicit about which other factors we account for and have strengthened our description with mathematical expressions for the determination of our volume and thickness errors (equations 2 and 3).

We also highlight our desire to further tackle the largest sources of uncertainty, and their associated errors, in our concluding paragraph. The relevant sentence reads:

*"The next steps in the advancement of the data are to develop improved estimates of snow loading on Arctic sea ice, and to* |

| | | measurements of sea ice thickness determined from airborne and in situ platforms. Were this not to be the case, we would agree with the assertion that our errors are not well characterised. However, it is, and so we believe that our error budget is in fact a reasonable and credible assessment of the uncertainty in our retrieval. The reviewer does make some specific suggestions as to how our error budget might be modified to suit the case of the near real time data set, which is spatially and temporally under-sampled relative to the archive product. We agree that these suggestions make good sense, and so we have modified our error budget to take these additional uncertainties into account.. | *further constrain the uncertainties in snow loading and sea ice density."* |
|---|---|---|---|
| **R2.2** | "In several areas of the text the mathematical operations performed on the data need to be explicitly written out as otherwise it is unclear exactly how some of the calculations were done. One example on this is that it is unclear whether a correction for the slower speed of light in snow has been applied to the calculation of freeboard. It is stated in Tilling et al., 2015 "A correction is applied to each freeboard measurement to account for the attenuation of the radar pulse as it passes through any snow cover on sea ice, where snow depth is based on a climatology." But this sentence is confusing as it could also apply to attenuation of energy through the snow, which in itself would not necessarily impact the freeboard determination. If this factor is applied, and whether it was | We agree that it is unclear in our original manuscript whether a correction for the slower speed of light in snow has been applied to the calculation of freeboard.

We also accept that the use of the word "attenuation" could cause confusion.

We agree also that it would be helpful to the reader if the factors included in our error budget were stated more clearly in the text.

However, we do not agree that the mathematical operations performed on the data should be written out in full, because they do not differ from those presented in an earlier manuscript (Tilling et al., 2015); the aim of this study is to merely apply our method to fast delivery CryoSat-2 data and compare to archive results. | We have added a sentence to the methods paragraph stating that:

*"A correction is applied to each freeboard measurement to account for the reduced speed of the radar pulse as it passes through any snow cover on sea ice."*

Should the reader require any further information on our methods we now direct them explicitly to Tilling *et al*. (2015). The second Data and Methods paragraph, first sentence, now reads:

*"The processing steps for fast delivery CryoSat-2 data are identical to those used for the final delivery data, and are described in Tilling et al. (2015)."*

We have also included a more in-depth description of our error analysis, and strengthened this with mathematical expressions for the determination of our volume and thickness errors (equations 2 and 3). |

| | | | |
|---|---|---|---|
| | applied in the determination of sea ice thickness and volume uncertainty, is not clear in the text." | | |
| **R2.3** | "It is also unclear how freeboard retrieval errors would propagate into the uncertainty calculations. Tilling et al., 2015 state that an interpolation is done between ocean surface elevation measurements to determine freeboard. The interpolation procedure was not explicitly stated but needs to be done so here. Any such interpolation would change the correlation length of the errors in the assessment and needs to be considered." | We agree that we should reconsider the contribution of freeboard uncertainty associated with the sparse sampling of the near real time products computed over short time intervals.

We do this by comparing sea surface height profiles along individual Arctic passes for crossovers where the time between the ascending and descending arc is sufficiently small that the real sea surface height has not varied significantly (say three days or less). On average, sea surface heights have a standard deviation of ~6 cm. When combined with the difference between the sea surface height of the ascending and descending arc, the total uncertainty on an individual interpolated sea surface height is ~4 cm. We interpolate sea surface heights using along-track linear regression with a moving window of width 200km, so this uncertainty contribution due to sea surface height interpolation will be correlated between freeboard measurements along the same satellite pass separated by 200 km or less.

We also agree with the reviewer that we should explicitly state the interpolation procedure. | We now consider the contribution of freeboard uncertainty, due to sea surface height interpolation, to our sea ice thickness error. This is considered separately to the contribution of freeboard uncertainty due to floe height measurement uncertainty, which is caused by radar speckle and random noise in the retracking step. Both of these are explained in detail in the text, with regards to their contribution to uncertainty in sea ice volume (third Data and Methods paragraph) and sea ice thickness (fourth Data and Methods paragraph).

The interpolation procedure is now explicitly stated in the text. The relevant sentence reads:

*"Sea surface height measurements are interpolated using along-track linear regression with a moving window of width 200km. "* |
| **R2.4** | **P2L25:** "The need for model ingestion is mentioned. But it should be considered that many models which ingest data have trouble with gridded mean sea ice thickness data and prefer to work with swath level data because sea ice thickness in modern models is represented as a distribution rather than a mean value. It | Although we acknowledge that different data formats may be desired by different users, we provide the gridded product as it is compact and evenly distributed, to satisfy a wide range of users. Bespoke products, such as swath level data, are available on request.

. | No changes made, because the remark relates to our data product rather than the manuscript. |

| | | | |
|---|---|---|---|
| | would be more useful to provide the point to point measurements of freeboard (the actual measurement made by CryoSat-2) which could be more easily ingested in a model." | | |
| R2.5 | **P4:** "The mathematical expression for determination of sea ice thickness error needs to be written out." | We agree that it would be helpful to the reader if we included the mathematical expression for the determination of sea ice thickness error. | We have included a mathematical expression for the conversion of sea ice freeboard to thickness (equation 1) to introduce the processing step at which the uncertainties are introduced. We have also expanded our description of our error analysis, and strengthened this with mathematical expressions for the determination of our errors (equations 2 and 3). |
| R2.6 | **P4:** "Was the uncertainty due to the lower speed of light in snow considered in the error estimates?" | We appreciate that this was not clear from the paper | We have included a more in-depth description of our error analysis, and strengthened this with mathematical expressions for the determination of our volume and thickness errors (equations 2 and 3). From this we hope that it is clear that the uncertainty due to the lower speed of light in snow was not considered in our error estimate. However, we have also included explicit reference to this in our concluding paragraph by stating that:

*"Our sea ice thickness and volume error budget could be further constrained by improved knowledge regarding the uncertainties in snow loading and sea ice density, as well as accounting for the uncertainty due to the reduced speed of light propagation through the snow pack."* |
| R2.7 | **P4L27:** "The mathematical expression for the circular operator needs to be written out as it is unclear how this was applied to the data." | We agree that we do not make it clear how the circular operator was applied to the data. On consideration, the phrase 'circular operator' is misleading and needs to be removed. | The relevant sentence now reads:

*"To obtain Arctic-wide and ROI grid values, we average all thickness measurements within a 25 and 5 km radius of the centre of the grid, respectively, with all points receiving equal weighting."* |

| | | | We have also removed the reference to the 'circular operator' in the caption of Figure 1. |
|---|---|---|---|
| R2.8 | **P3L19:** "The reference to Kwok et al., 2009 is confusing here as the paper does not describe the use of CryoSat-2 data." | We agree that the reference to Kwok *et al.*, 2009 is confusing, and that we need to clarify how it is relevant to CryoSat-2 data | The sentence now reads: *"NASA provide monthly-averaged thickness data for March 2014 and March 2015 within a fixed central Arctic region that covers an area of ~7.2×10$^6$ km$^2$. The region was first defined for use with the NASA ICESat satellite [Kwok et al., 2009], and will herein be referred to as the ICESat domain."* |
| R2.9 | **P4L5:** "Which geophysical corrections are often missing in the data? They should be listed." | We agree that the geophysical corrections should be listed | The sentence has been expanded to read: *"In the fast delivery data the wet tropospheric, dry tropospheric and inverse barometer corrections are missing in 93.8% of cases for Baseline-B data, but only 0.3% of cases for Baseline-C data. In these cases, all three of the corrections are missing. "* We have moved the sentence further up in the paragraph as we feel it makes more sense to include it immediately after the baseline processing is introduced. |
| R2.10 | **P4L16-17:** "How is snow from the Warren climatology applied beyond areas of the central Arctic? The reasons for this were mentioned clearly in the other review. I think this is a critical part of the manuscript as this could have a large impact on first year ice areas outside of the central Arctic basin." | We appreciate the referee's concern regarding the Warren climatology, especially in regions where it is not constrained by *in situ* measurements. To avoid using unconstrained value of snow depth and snow density we use the mean climatology values of snow loading from a fixed central Arctic domain (where snow parameters are constrained) in all freeboard to thickness conversions, no matter where they are located. There are known differences between the climatology and the current snow depth on younger Arctic sea ice (Kurtz *et al*. 2011; Webster *et al*. 2014) so we halve the snow depth on FYI to account for reduced snow accumulation. This should be explicitly stated in the paper. | A sentence has been added to summarise our treatment of the Warren climatology. It reads: *"To obtain snow depth and density we average the values from a climatology (Warren et al. 1999) that fall within the ICESat domain, where the climatology is constrained by in situ measurements."* The ICESat domain is defined earlier in the paper. Should the reader require further information, the second paragraph in the Data and Methods section, first sentence, now reads: |

| | | | "*The processing steps for fast delivery CryoSat-2 data are identical to those used for the final delivery data, and are described in Tilling et al. (2015).*" |
|---|---|---|---|
| R2.11 | **P4L17-18:** "The specific densities for sea ice and water need to be written out." | We agree that these densities should be written out | We have added the densities to the paper. The relevant sentence now reads: "We use a fixed estimate of first-year ice (FYI) density of 916.7 kg m$^{-3}$, multi-year ice (MYI) density of 882 kg m$^{-3}$ [*Alexandrov et al.*, 2010], and a fixed seawater density of 1,023.8 kg m$^{-3}$ [*Wadhams et al.*, 1992]." |
| R2.12 | **P4L26:** "If a 1 km grid can be provided, why not also provide the swath level freeboard data which is of similar resolution?" | We appreciate that some users would prefer to have swath level data.

However, this paper is intended as an introduction to the dataset that is currently publicly available. We provide the gridded product as it is compacts and evenly distributed, to satisfy a wide range of users. The 1km data is available over reduced regions of interest, so is still more compact than numerous satellite swaths. Bespoke products, such as swath level data, are available for collaborators on request. | No changes made, because the remark relates to our data product rather than the manuscript. |
| R2.13 | **P4L37:** "Given the extrapolations of the Warren climatology outside of the central Arctic, as well as the modified version over first year ice, I would question these snow depth uncertainty estimates as they have been quite modified from their original source." | We agree with the referee that snow depth has been quite modified from its original source, and that this may cause issues with uncertainty estimates.

However, there is a lack of real knowledge regarding the uncertainties in snow depth, as well as snow density, and sea ice density. We have attempted to account for this lack of knowledge in our error budget by including errors of snow depth, snow density and sea ice density that are likely an overestimate, owing to the sparse spatial and temporal sampling of the measurements [*Tilling et al.*, 2015]. We have developed the most comprehensive error budget we can, considering this lack of knowledge. We | We now highlight our desire to tackle this issue in our concluding paragraph. The relevant sentence reads:

"*The next steps in the advancement of the data are to develop improved estimates of snow loading on Arctic sea ice, and to further constrain the uncertainties in snow loading and sea ice density.*" |

| | | believe that our error budget is a reasonable estimate of uncertainty, as the values are consistent with published comparisons of CryoSat-2 sea ice thickness estimates with independent measurements of thickness and draft from airborne and ocean-based platforms (Tilling *et al.*, 2015). | |
|---|---|---|---|
| R2.14 | **P4L42-44:** "The statement that the large number of freeboard measurements negates the uncertainty rests on the assumption that the errors are uncorrelated in space and time. This seems highly unlikely given that the retrieval method does not account for factors such as changing snow conditions as shown by Ricker et al., 2015." | We do not assume that uncertainties in freeboard are uncorrelated in space and time, as the referee suggests. Rather, we have attempted to characterise the degree to which they are correlated using an empirically determined length scale within our error budget. This approach leads to larger uncertainties when compared to error budgets that assume uncorrelated uncertainties (e.g. Ricker *et al.*, 2014).

Again, our error model leads to uncertainties in sea ice thickness that are comparable to the spread of differences from independent measurements determined from airborne and in situ platforms, and this leads us to believe that the model is in fact a reasonable and credible assessment of the uncertainty in our retrieval. | No changes made |
| R2.15 | **P5L1-7:** "The method for determining volume uncertainties is unclear and should be written out mathematically to fully describe the procedure. Also, over what range is each parameter adjusted to calculate the rate of change?" | We agree that it would be helpful to include the mathematical expression for the determination of sea ice volume error | We have included a more in-depth description of our error analysis, and strengthened this with mathematical expressions for the determination of our volume and thickness errors (equations 2 and 3). |

| | | | |
|---|---|---|---|
| **R2.16** | **P5 second paragraph:** "I think this estimate of error is a gross simplification of the un- certainties and is not accurate. For the snow depth term, it was already acknowledged that there are large differences over first year and multi-year ice which are unrelated to synoptic scale meteorology but is rather related to the timing of snow fall events and ice freeze-up. Sea ice density would also similarly be unrelated to synoptic scale meteorology particularly as the values used in the study are based on first year and multi-year ice types. I would therefore not consider the 2000 km decorrelation length to be accurate. Have you looked at other data to determine the decorrelation length for these parameters?" | We appreciate the referee's concern that our estimate of error is a simplification. However, there is a lack of real knowledge regarding the uncertainties in snow depth, snow density, and sea ice density. We have attempted to account for this lack of knowledge in our error budget by including errors of snow depth, snow density and sea ice density that are likely an overestimate, owing to the sparse spatial and temporal sampling of the measurements [*Tilling et al.*, 2015]. We have developed the most comprehensive error budget we can considering this lack of knowledge. Our uncertainty estimates are consistent with published comparisons of CryoSat-2 sea ice thickness estimates with independent measurements of thickness and draft from airborne and ocean-based platforms (Tilling *et al.*, 2015).

Again, we believe that attempting to characterise a de-correlation length scale is an improvement on alternative error budgets that assume uncorrelated uncertainties. | We have expanded our error budget to include the contribution of sea ice freeboard uncertainty due to a.) sea surface height uncertainty and b.) floe height measurement uncertainty (due to radar speckle and random noise in the retracking step). Please see the response and action to **R2.3** and **R2.17**.

We have added the treatment of a.) and b.) to the description of our error analysis, and introduce these by stating that

"*The construction of our error budget is described in Tilling et al. [2015], but we now expand on this by considering the contribution of uncertainty in sea ice freeboard in more detail.*"

We have been explicit about which other factors we account for and have strengthened our description with mathematical expressions for the determination of our volume and thickness errors (equations 2 and 3).

We now take care to be completely transparent about the difficulties associated with determining de-correlation lengths for contributing uncertainty factors. We open the fourth Data and Methods paragraph by saying:

"*Estimating the error on individual or grid cell sea ice thickness measurements is complicated by lack of knowledge regarding the de-correlation length scales of the contributing uncertainty factors.*" |
| **R2.17** | **P5 second paragraph:** "The last sentence in this paragraph not accurate as there is likely residual error in the sea surface | We agree with the referee that it is important to consider how spatial variations in sea surface height references will impact on sea ice thickness uncertainty. | We now consider the impact of spatial variations in sea surface height references (when calculating sea ice freeboard) on sea ice thickness uncertainty. This is |

| | | | |
|---|---|---|---|
| | height estimate since there is a need to interpolate over data gaps due to the varying number of lead points available. The interpolation procedure needs to be written out so that the correlation length of errors in the sea ice thickness can be better understood and taken into account." | Although uncertainty in sea surface height (∼4 cm, see response to **R2.3**) will be a negligible component of our monthly volume uncertainty, as we typically include more than 1 million floe heights and 10,000 200 km arc segments when computing, it will impact on thickness uncertainty as the sea surface height uncertainty will remain correlated along each satellite pass crossing a 25 km radius averaging window. We estimate that the effect will be reduced in the averaging only by the square root of the number of individual passes crossing a significant part of the averaging window. Therefore the impact of sea surface height uncertainty on the overall thickness error budget will have a greater impact on thickness for shorter timescales and at lower latitudes, due to the increased sparsity in spatial sampling.

We also agree that the interpolation procedure needs to be written out. | described in detail in the text (fourth Data and Methods paragraph). We explain that the magnitude of the contribution of sea surface height uncertainty to our thickness error budget depends on the spatial sampling of the data. We back this up by including typical values for the total thickness uncertainty for varying degrees of data coverage.

The interpolation procedure is now written out in the text. The relevant sentence reads:

"*Sea surface height measurements are interpolated using along-track linear regression with a moving window of width 200km.*"

We now take great care to be completely transparent about the difficulties associated with determining de-correlation lengths for contributing uncertainty factors. We open the fourth Data and Methods paragraph by saying:

"*Estimating the error on individual or grid cell sea ice thickness measurements is complicated by lack of knowledge regarding the de-correlation length scales of the contributing uncertainty factors.*" |
| **R2.18** | **Figure 2a:** "There appear to be negative ice thickness values in the distribution, I'm guessing this is due to uncertainties in the freeboard retrieval but some explanation on this is in order." | The referee is correct that negative ice thickness values are due to uncertainties in the freeboard retrieval. We agree that some explanation is necessary. | We have added a sentence that reads:

"*The negative thickness values apparent in Figures 2a and 2b are a consequence of negative freeboard measurements that occur due to random noise in the returns from thin ice floes, caused by radar speckle. These freeboards are included in our processing to ensure that the average freeboard, and therefore thickness, is not biased high.*" |
| **R2.19** | "A map of the differences with the final data compared to the NRT also needs to be shown. This will reveal whether | We understand that readers may desire more information with regards to the spatial differences between NRT and archive sea ice thickness products. | We have included a new figure (Figure 3), which consists of 2 maps, detailing the spatial differences between NRT and archive sea ice thickness data for absent and present |

| | | |
|---|---|---|
| | regional differences are present." | geophysical corrections. The explanatory text for this figure (Data and Methods final paragraph, final few sentences) reads: |
| | Since our initial submission we have found that the absence of certain geophysical corrections (wet tropospheric, dry tropospheric and inverse barometer), caused the most noticeable differences in NRT and archive sea ice thickness. We feel that the best way to display this is by plotting the spatial variability of these differences for two different months: one with corrections absent and one with corrections present. | "*The remaining difference is likely due to the combined absence of the wet tropospheric, dry tropospheric and inverse barometer corrections in 93.8% of the Baseline-B fast delivery CryoSat-2 data. This is reduced to 0.3% for Baseline-C data. The mean sea ice thickness for both the NRT and archive datasets is ~1.8 m, and there is no bias between them, with or without geophysical corrections applied. When the corrections are missing the NRT and archive thickness values at any given location differ, on average, by 1.1 cm with a standard deviation of 23.0 cm (Figure 3a). This is reduced to 0.1 cm with a standard deviation of 7.4 cm when the corrections are present (Figure 3b). There is no spatial pattern to these differences. Despite the improvement in performance of Baseline-C NRT data compared with Baseline-B we conclude that the satellite orbits and on-ground processing applied to fast delivery CryoSat-2 data are sufficient to determine accurate measurements of Arctic sea ice thickness and volume for both baselines. The thickness differences between the archive and NRT data products are not significant for either baseline given the estimated uncertainty on thickness and the typical thickness of sea ice floes.*" |

**References**

Alexandrov, V., S. Sandven, J. Wahlin, and O. M. Johannessen (2010), The relation between sea ice thickness and freeboard in the Arctic, *The Cryosphere*, *4*, 373-380.

Kurtz, N. T., N. Galin, and M. Studinger (2014), An improved CryoSat-2 sea ice freeboard retrieval algorithm through the use of waveform fitting, *The Cryosphere*, *8*, 1217-1237.

Kwok, R., G. F. Cunningham, M. Wensnahan, I. Rigor, H. J. Zwally, and D. Yi (2009), Thinning and volume loss of the Arctic Ocean sea ice cover: 2003-2008, *Journal of Geophysical Research-Oceans*, *114*(C7), C07005-07001 - C07005-07016.

Ricker, R., S. Hendricks, V. Helm, H. Skourup, and M. Davidson (2014), Sensitivity of CryoSat-2 Arctic sea-ice freeboard and thickness on radar-waveform interpretation, *The Cryosphere*, *8*(4), 1607-1622.

Tilling, R. L., A. Ridout, A. Shepherd, and D. J. Wingham (2015), Increased Arctic sea ice volume after anomalously low melting in 2013, *Nature Geoscience*, *8*, 643-646.

Wadhams, P., W. B. I. Tucker, W. B. Krabill, J. C. Swift, J. C. Comiso, and N. R. Davis (1992), Relationship between sea ice freeboard and draft in the Arctic Basin, and implications for ice thickness monitoring, *Journal of Geophysical Research-Oceans*, *97*(C12), 20325–20334.

---

## Author Response (AR2)

**Response to Editor's comments**

Below we summarise the comments of the Editor, along with our responses and actions:

| # | Comment (verbatim) | Response | Action |
|---|---|---|---|
| **E1** | "In most cases it is hard to see what the scientific benefits of NRT data really are. I think here is your chance to give that product and yourself a little more scientific credibility by providing a more critical view." | We appreciate that the manuscript would benefit from us being more specific and critical about the science benefits of our NRT data provision.

Currently the leading scientific benefit of NRT sea ice thickness observations is in the seeding of short-term forecast models, such as the U.S Navy's Arctic Cap Nowcast/Forecast System (ACNFS) (Hebert et al., 2015, Posey et al., 2011), which provides forecasts on a 1-7 day timescale. The developers of the Centre National de Recherches Meteorologiques Coupled Global Climate Model, version 3.3 (CNRM-CM3.3), which is a seasonal forecast model, have also suggested that September sea ice extent is potentially predictable up to 6 months in advance if accurate observations of sea ice thickness are available (Chevallier and Salas-Melia, 2012). However, the usefulness of sea ice thickness data from the month of May or before for September sea ice prediction is still under contention (Day et al., 2014). | We have re-written our introduction, discussion and conclusions to be explicit and critical about the scientific and operational benefits and restrictions of our NRT sea ice thickness product. |
| **E2** | "So what are NRT CryoSat data good for? And what do you mean by operational users? An evaluation needs to consider the accuracy and spatial and time resolution of the product. I would argue that one thickness retrieval every 14 or 6 km isn't at all enough to help tactical (short-term) marine operations, and thus | We agree that we did not fully explain the limits on the usefulness of our data for operational users. However, operational users may benefit from the output of operational model predictions, which currently assimilate sea ice concentration data and could be improved by the assimilation of NRT thickness data. However, whether or not ice thickness data from before May will improve summer predictions is still a matter of contention. | We have removed the dialogue about operational users from our first introductory paragraph and from our discussion and conclusions. They have been re-written to be more critical of the importance of NRT sea ice thickness data for operational use. Specifically, we now concentrate on the assimilation of NRT ice thickness data into operational models. We also highlight the limitations of our data, as they are not available during the summer. |

| | | | |
|---|---|---|---|
| | the product will not be useful for planning ship routes or drilling operations." | | |
| **E3** | "In addition, you should specify what the CryoSat retrievals actually represent. Is it mean thickness, modal thickness, or maximum thickness? There seems to be some consent that it is modal thickness. However, for shipping and ice management it is the tail of the thickness distribution and the amount and thickness of the thickest ice that are of most concern." | We agree that the question of what CryoSat retrievals represent is an important one for operational users in the Arctic. However, this still the topic of some debate, between ourselves and others in the sea ice community. If the Editor knows of some work that addresses this question then we would be very grateful to be directed to it. | As we cannot definitively answer the question of what CryoSat retrievals represent, we have toned down our claims of the usefulness of the data for operational users. Please see action to **E1** and **E2**. |
| **E4** | "Marine operations are mostly carried out in the summer. That is the time when your product is not available. Also you talk about the Northwest Passage a lot, but only mention briefly that the data are of lesser quality there. A more critical discussion would be desirable." | We agree that operational users would benefit from NRT ice thickness data in summer months for the assimilation into operational forecast models. However, some operational model outputs, such as that from the ACNFS, are used year-round. For seasonal forecast models the usefulness of sea ice thickness data from the months of May or before is still a matter of contention. | In our introduction we now discuss the operational implications, with regards to operational models, of NRT data provision. Included are the limitations of the temporal coverage of our NRT data. This is repeated in the discussion and conclusions of the manuscript. |
| **E5** | "On the other end of the scale, you argue that the NRT data will improve climate models etc. I cannot see why such long-term activities and developments, and the observation of climate related changes would benefit from NRT data and couldn't simply use the traditional products?" | We agree that long-term climate modelling is not likely to benefit from the inclusion of NRT sea ice thickness data over traditional products. This was an oversight on our part; our reference to 'climate models' should have emphasised when they are used for short-term forecasting. This is summarised in our response to **E1**. | Please see action to **E1** and **E2**. |
| **E6** | "Also what is the advantage of NRT ice volume estimates (Arctic wide?) over more regional ice thickness information [for climate models]?" | We appreciate that we need to be more specific regarding the benefits of NRT sea ice volume data compared with thickness data. Experiments with the CNRM-CM3.3 seasonal forecast model have shown that there is a higher potential to predict the September ice area by using the | We have included a sentence related to this in our first introductory paragraph, and have been explicit regarding the temporal limitations of our data. The relevant sentences read:
"Despite these potential benefits, it is nevertheless |

| | | | |
|---|---|---|---|
| | | sea ice volume anomaly rather than thickness or concentration. However, this is only true for the month of June, when our data is not available (Chevallier and Salas-Melia, 2012).

Another advantage is that we can report NRT data to the scientific community in a timely way, which is important for those wishing to communicate the state of the Arctic beyond the scientific community. | recognized that the value of NRT sea ice thickness observations derived from repeat satellite altimetry does have limits. For example, some model systems show higher forecast skill when initialized with thickness (and for some months volume) observations acquired during early summer (Chevallier and Salas-Melia, 2012). Summer is a period when sea ice thickness measurements are traditionally unavailable in the Arctic due to the presence of melt ponds (e.g. Tilling et al., 2015)."

We have discussed the importance of our data to the scientific community in the second introductory paragraph. |
| **E7** | "Finally, I do think that the one and only aspect where NRT data even with monthly resolution could be useful is seasonal ice forecasting, where observed growth during the winter and ice thickness in the end of the winter could be used to evaluate the state and pre-conditioning of the ice cover in a certain year, and where late spring ice thickness data can be used to initialize forecast models to inform outlooks of general summer ice conditions. This could be helpful for e.g. strategic (long-term) navigation planning (go or not go...). This is also one of the objectives of OIB, but the CryoSat product may be more useful than the OIB data due to their larger regional coverage, at the cost of much smaller spatial resolution. Lindsay et al. (2012), for example, have shown how such data can be used for ice forecasting, and it would only be natural to suggest to do the same with CryoSat data. | We agree with the Editor that the key application of the NRT data is in seasonal ice forecasting, as well a short-term (1-7 day) forecasts. We did not make this clear in the Introduction or Discussion of our manuscript and instead suggested that longer-term climate models could benefit from the inclusion of NRT data. We now appreciate that this is not necessarily the case. Please see response to **E2**. However, we believe that our NRT data is important for timely assessments of the state of the Arctic, which is a key responsibility for science.

We agree that the paper should end on a more critical note than it currently does. | Please see action to **E1** and **E2**.

We feel that the manuscript now ends on a more critical note. |

| | | | |
|---|---|---|---|
| | I think it would be good if you could end your paper on such a note…” | | |

**References**

[revised manuscript text omitted]